# On the Design of a New Simulated Inductor Using a Contactless Electrical Tomography System as an Example

**DOI:** 10.3390/s19112463

**Published:** 2019-05-29

**Authors:** Xin Ye, Yuxin Wang, Xiao-Yu Tang, Haifeng Ji, Baoliang Wang, Zhiyao Huang

**Affiliations:** State Key Laboratory of Industrial Control Technology, College of Control Science and Engineering, Zhejiang University, Hangzhou 310027, China; yexin0601@zju.edu.cn (X.Y.); wangyx2015@zju.edu.cn (Y.W.); xytang@zju.edu.cn (X.-Y.T.); hfji@zju.edu.cn (H.J.); wangbl@zju.edu.cn (B.W.)

**Keywords:** simulated inductor technique, process tomography, contactless electrical tomography

## Abstract

This work reports a new simulated inductor which is suitable for a Contactless Electrical Tomography (CET) system and can effectively overcome the unfavorable influence of coupling capacitance on the measurement results. By detailed analysis and comparison, it is found that the grounded simulated inductor has a simple circuit construction but its output current is not equal to its input current, while the floating simulated inductor can be used as an independent inductor module but its circuit structure is relatively complex. A new simulated inductor is designed by compensating the currents from the common node of an introduced independent power source to the main circuit. The new simulated inductor combines the advantages of the grounded simulated inductor and the floating simulated inductor. It has the simple construction similar to that of the grounded simulated inductor and its input current is equal to the output current, which means it can be used as an independent module. The impedance measurement and practical image reconstruction experiments were carried out to verify the effectiveness of the new simulated inductor. The experimental results show that the design of the new simulated inductor is successful, and the performance of the impedance measurement is satisfactory. The signal-to-noise ratio of the CET system is improved. Meanwhile, the research work also indicates that in the case when the independent power source is not available, the new simulated inductor is also an effective alternative method. But the phase difference between input signal and output signal is approximately 90° when the elimination principle is realized.

## 1. Introduction

As basic electromagnetic devices, inductors are widely used in circuit design and signal processing [1,2,3]. Inductance is the circuit parameter used to describe an inductor, which relates the induced voltage to the current. However, the conventional practical inductors are usually made up of coils and magnetic cores. It is always difficult to implement a practical inductor with large inductance value and small physical size. Besides, the inductance value of a practical inductor can’t be adjusted easily. Even for the practical adjustable inductor, its adjustment range is usually less than 15%.

The simulated inductor technique was developed and studied to replace a practical inductor effectively in the research field of integrated circuits [3,4,5,6,7,8,9,10,11,12,13,14]. A simulated inductor is an active circuit for generating an equivalent inductive reactance, which is implemented with active and passive components (such as resistors, capacitors and operational a mplifiers) [6]. Compared with a practical inductor, a simulated inductor has comparable function performance. It also has the advantages of easily implementing a larger inductance value and acquiring a wider adjustable range of inductance. Meanwhile, because the miniaturization of the components (resistors, capacitors and operational amplifiers) used in simulated inductors is easy to realize at the current technique level, it is easy to achieve miniaturization of simulated inductors.

It is worth mentioning that the simulated inductor has high skill requirements for the circuit design, although it has significant advantages over practical inductors [4]. The simulated inductors can be roughly divided into two types: the grounded simulated inductor and the floating simulated inductor [10,11,12,13,14,15]. The grounded simulated inductor has the advantage of a simple circuit structure, but one of its terminals needs to be grounded directly. The floating simulated inductor does not have to be grounded directly, and it can be regarded as an independent module and be connected to the required position of an application circuit, but it has the disadvantages of a complicated circuit structure and high component matching requirements. These design requirements of the two simulated inductors more or less limit the practical applications of the simulated inductor technique. Up to date, the simulated inductor technique has mainly been studied and used as part of the active filters in the field of electronic communication. Few research works on the application of the simulated inductor technique in other research fields are reported. Therefore, it will be of great significance if we can seek an effective approach to develop a new kind of simulated inductor which combines the advantages of the two conventional simulated inductors mentioned above and overcomes the existing disadvantages of each type. That would extend the application fields of the simulated inductor technique and satisfy the wide requirements of an inductor (or inductor module) with large inductance value, wide adjustable range of inductance value and small size.

Currently, contactless electrical tomography (CET) has received increasing attention in the field of process tomography [16,17]. The latest research progresses have displayed the potential and broad perspective of the contactless electrical tomography (CET) techniques [18,19,20,21,22,23,24,25,26,27]. However, the current CET systems still exist a drawback, i.e., the existence of the coupling capacitances (formed by the two electrodes, insulating pipe, and the measured fluid) are adverse to the detection of the useful measurement signal (the details will be discussed in Section 5). Research works have verified that the impedance elimination principle can provide an effective approach [28,29,30,31,32], i.e., by introducing an inductor module, using the inductive reactance of the inductor module to eliminate the capacitive reactance of the coupling capacitances. However, that needs large-value and adjustable independent inductors. The emergence of simulated inductor technique provides an effective approach to solve this problem. Unfortunately, at the current stage, our knowledge and experience on applying simulated inductor technique to the field of process tomography is limited. More research work needs to be undertaken.

This work aims to report a new simulated inductor which is suitable for CET systems. The new simulated inductor, which is designed on the basis of the classic Riordan circuit, has a simple circuit structure and can be regarded as an independent module. The research work mainly includes the following four parts:(1)The analysis and discussions on the characteristics of the Riordan simulated inductors.(2)The requirements of the inductor module in CET system.(3)The design of the new simulated inductor module.(4)The experimental results with the new simulated inductor.

## 2. Riordan Simulated Inductor

Simulated inductors based on the Riordan circuit are widely accepted and studied by many scholars [3,4]. The Riordan simulated inductors can also be divided into two types, the Riordan grounded simulated inductor and the Riordan floating simulated inductor. Figure 1a,b show the typical circuits of a Riordan grounded simulated inductor and a Riordan floating simulated inductor, respectively. For both Riordan simulated inductors, the circuits input impedance $Z_{in}$ both behave as an equivalent inductance $L_{\mathrm{eq}}$, i.e.,:(1)$$Z_{in}=\frac{u_{in}}{i_{in}}=j2\pi fL_{eq}$$
where f is the frequency of the excitation signal and j is the imaginary unit.

The Riordan grounded simulated inductor has a simple circuit structure, as shown in Figure 1a. One of its terminals must be grounded directly. The Riordan floating simulated inductor uses two Riordan grounded simulated inductors which are connected in cascade (back to back), as shown in Figure 1b. It has a symmetrical circuit structure. Its terminals need not to be grounded directly. It can be regarded as an independent module (like a practical inductor) and be connected to the required position of an electrical circuit freely.

The detailed discussions and analysis on the characteristics of the Riordan grounded simulated inductor and the Riordan floating simulated inductor are given in the following sections.

### 2.1. The Characteristics of the Riordan Grounded Simulated Inductor

Let $u_{in}$ be the input voltage, $i_{in}$ be the input current, $i_{out}$ be the output current, $i_{A1}$ and $i_{A2}$ be the current flowing into the operational amplifier $A_{1}$ and $A_{2}$ respectively. The following equations can be obtained:(2)$$i_{in}=i_{A2}+i_{R2}=i_{A2}+i_{A1}+i_{out}$$
(3)$$u_{A2}=u_{in}-i_{in}R_{1}$$
(4)$$i_{R2}=\frac{u_{A2}-u_{in}}{R_{2}}=-\frac{R_{1}}{R_{2}}i_{in}$$
(5)$$i_{A2}=i_{in}-i_{R2}=\frac{R_{1}+R_{2}}{R_{2}}i_{in}$$
(6)$$u_{A1}=u_{in}-i_{R2}R_{3}=u_{in}+\frac{R_{1}R_{3}}{R_{2}}i_{in}$$
(7)$$i_{out}=\frac{u_{A1}-u_{in}}{\frac{1}{1/R_{4}+j2\pi fC_{m}}}=\frac{\frac{R_{1}R_{3}}{R_{2}}i_{in}}{\frac{1}{1/R_{4}+j2\pi fC_{m}}}=\frac{\left(1/R_{4}+j2\pi fC_{m}\right)R_{1}R_{3}}{R_{2}}i_{in}$$
(8)$$i_{A1}=i_{R2}-i_{out}=-\frac{R_{1}+\left(1/R_{4}+j2\pi fC_{m}\right)R_{1}R_{3}}{R_{2}}i_{in}$$

Meanwhile, the output current $i_{out}$ can also be described as:(9)$$i_{out}=i_{R5}=\frac{u_{in}}{R_{5}}$$

According to Equations (7) and (9), input impedance of the Riordan grounded simulated inductor $Z_{in}$ is:(10)$$Z_{in}=\frac{u_{in}}{i_{in}}=\frac{\left(1/R_{4}+j2\pi fC_{m}\right)R_{1}R_{3}R_{5}}{R_{2}}=\frac{R_{1}R_{3}R_{5}}{R_{2}R_{4}}+j2\pi f\frac{C_{m}R_{1}R_{3}R_{5}}{R_{2}}$$

Thus, the equivalent inductance *L_eq_* of the Riordan grounded simulated inductor is:(11)$$L_{eq}=\frac{C_{m}R_{3}R_{5}}{R_{2}}R_{1}$$

The equivalent internal resistance $r_{eq}$ of the Riordan grounded simulated inductor is:(12)$$r_{eq}=\frac{R_{3}R_{5}}{R_{2}R_{4}}R_{1}$$

According to Equations (11) and (12), the equivalent inductance $L_{eq}$ is determined by the resistances $R_{1}$
$R_{2}$, $R_{3}$, $R_{5}$ and the capacitance $C_{m}$, and the equivalent internal resistance $r_{eq}$ is determined by the resistances $R_{1}$
$R_{2}$, $R_{3}$, $R_{4}$ and $R_{5}$. Usually, in practical applications, in order to adjust the value of $L_{eq}$ and $r_{\mathrm{eq}}$ conveniently and independently, the values of $R_{2}$, $R_{3}$, $R_{5}$ and $C_{m}$ are fixed. The value of $L_{eq}$ is mainly adjusted by changing the value of the high-precision adjustable resistor $R_{1}$, while the value of $r_{\mathrm{eq}}$. is mainly determined by the value of $R_{4}$. Because the equivalent internal resistance of an inductor module should be as small as possible, the value of $R_{4}$ is usually much larger than those of $R_{1}$, $R_{2}$, $R_{3}$ and $R_{5}$, respectively.

It is necessary to indicate that the value of the $L_{eq}$ is affected by $R_{5}$. In practical applications, the value of $L_{eq}$ is expected as a constant, which means the value of $R_{5}$ should also be a constant. That is the reason why one terminal of the circuit is grounded. Otherwise, it is difficult to guarantee the value of $L_{eq}$ to be a constant and the value of $L_{eq}$ will vary with the impedance of the succeeding circuit. In other words, $R_{5}$ could be regarded as the load of the circuit. If the output terminal is not grounded, the load of the circuit will change, and then $L_{eq}$ can not be fixed.

Meanwhile, Equation (7) also shows that the input current $i_{in}$ is not equal to the output current $i_{out}$, (i.e., $i_{in}\neq i_{out}$) and there exists a phase difference (approximately 90°) between $i_{in}$ and $i_{out}$. From Equations (2), (5) and (8), it can be found that there are currents flowing through the output of the operational amplifiers ($A_{1}$ and $A_{2}$), leading to the difference between $i_{in}$ and $i_{out}$.

Based on the discussions above, it can be found that although the Riordan grounded simulated inductor has a simple circuit construction, it has the disadvantage that it can’t constitute an independent module like the practical inductor yet. The reason is that the currents through the two terminals of the inductor module are not equivalent to each other.

### 2.2. The Characteristics of the Riordan Floating Simulated Inductor

As shown in Figure 1b, the Riordan floating simulated inductor has a symmetrical circuit structure. It is a combination of two identical Riordan grounded simulated inductors, i.e., the values of the resistances and capacitances of both sides are equal, i.e., $R_{1}=R_{1}{}^{\prime }$, $R_{2}=R_{2}{}^{\prime }$, $R_{3}=R_{3}{}^{\prime }$, $R_{4}=R_{4}{}^{\prime }$, $C_{m}=C_{m}{}^{\prime }$. Based on Equation (7), the current flow through $R_{5}$ is:(13)$$i_{R5}=\frac{\left(1/R_{4}+j2\pi fC_{m}\right)R_{1}R_{3}}{R_{2}}i_{in}$$

Further:(14)$$u_{out}=u_{in}-i_{R5}R_{5}$$
(15)$$u_{A3}=u_{out}-i_{R5}\frac{1}{1/R_{4}{}^{\prime }+j2\pi fC_{m}}=\frac{1/R_{4}+j2\pi fC_{m}}{1/R_{4}{}^{\prime }+j2\pi fC_{m}}\frac{R_{1}R_{3}}{R_{2}}i_{in}$$
(16)$$i_{R2}{}^{\prime }=\frac{u_{A3}-u_{out}}{R_{3}{}^{\prime }}$$
(17)$$u_{A4}=u_{out}-i_{R2}{}^{\prime }R_{2}{}^{\prime }$$
(18)$$i_{out}=\frac{u_{A4}-u_{out}}{R_{1}{}^{\prime }}=-\frac{R_{2}{}^{\prime }}{R_{1}{}^{\prime }}i_{R2}{}^{\prime }$$

According to Equations (13) and (14), the input impedance $Z_{in}$ of the Riordan floating simulated inductor can be described as:(19)$$Z_{in}=\frac{u_{in}-u_{out}}{i_{in}}=\frac{i_{R5}R_{5}}{i_{in}}=\frac{R_{1}R_{3}R_{5}}{R_{2}R_{4}}+j2\pi f\frac{C_{m}R_{1}R_{3}R_{5}}{R_{2}}$$

Thus, the equivalent inductance and the internal resistance $r_{eq}$ of the Riordan floating simulated inductor can be expressed as:(20)$$L_{eq}=\frac{C_{m}R_{1}R_{3}}{R_{2}}R_{5}$$
(21)$$r_{eq}=\frac{R_{1}R_{3}}{R_{2}R_{4}}R_{5}.$$

By comparison of Equations (11)–(12) and (20)–(21), respectively, we can see that the Riordan grounded simulated inductor and the Riordan floating simulated inductor have the same equivalent inductance and the same equivalent internal resistance. However, the way to adjust the value of $L_{eq}$ of the Riordan floating simulated inductor is different from the Riordan grounded simulated inductor. In practical applications, the value of $L_{eq}$ of the Riordan floating simulated inductor is mainly adjusted by changing the value of the adjustable resistor $R_{5}$. The value of $r_{eq}$ of the Riordan floating simulated inductor is mainly determined by the value of $R_{4}$.

Meanwhile, according to Equations (15) to (19), the output impedance $Z_{out}$ of the Riordan floating simulated inductor can be described as:(22)$$Z_{out}=\frac{u_{in}-u_{out}}{i_{out}}=\frac{i_{R5}R_{5}}{-\frac{R_{2}{}^{\prime }}{R_{1}{}^{\prime }}\left(\frac{-i_{R5}\frac{1}{1/R_{4}{}^{\prime }+j2\pi fC_{m}}}{R_{3}{}^{\prime }}\right)}=\frac{R_{1}{}^{\prime }R_{3}{}^{\prime }R_{5}}{R_{2}{}^{\prime }R_{4}{}^{\prime }}+j2\pi f\frac{C_{m}{}^{\prime }R_{1}{}^{\prime }R_{3}{}^{\prime }R_{5}}{R_{2}{}^{\prime }}$$

Because the Riordan floating simulated inductor is formed by two mirrorly-connected identical Riordan grounded simulated inductor. According to Equations (19) and (22), the input impedance $Z_{in}$ of the Riordan floating simulated inductor is equal to the output impedance $Z_{out}$, i.e.,(23)$$Z_{in}=Z_{out}$$

Further, from Equations (15), (16) and (18):(24)$$\begin{array}{c} i_{out}=-\frac{R_{2}{}^{\prime }}{R_{1}{}^{\prime }}i_{R2}^{{}^{\prime }}\\ =-\frac{R_{2}{}^{\prime }}{R_{1}{}^{\prime }}\left(\frac{u_{A3}-u_{out}}{R_{3}{}^{\prime }}\right)\\ =-\frac{R_{2}{}^{\prime }}{R_{1}{}^{\prime }}\left(-\frac{1/R_{4}+j2\pi fC_{m}}{1/R_{4}{}^{\prime }+j2\pi fC_{m}}\frac{R_{1}R_{3}}{R_{2}}i_{in}\right)\frac{1}{R_{3}{}^{\prime }}\\ =\frac{R_{2}{}^{\prime }R_{1}R_{3}}{R_{2}R_{1}{}^{\prime }R_{3}{}^{\prime }}i_{in}\\ =i_{in} \end{array}$$
i.e., the input current $i_{in}$ is equal to the output current $i_{out}$ and there is no phase difference between $i_{in}$ and $i_{out}$. Besides, the relationship between the currents flowing into the operational amplifiers $A_{3}$ and $A_{4}$ can be described respectively as:(25)$$i_{A3}=i_{R5}-i_{R2}{}^{\prime }$$
(26)$$i_{A4}=i_{R2}{}^{\prime }-i_{out}$$

According to Equations (5), (8), (14), (22)–(24), it can be found that the sum of the total current flowing into the output terminals of four operational amplifiers is zero, which means that the current flowing into the operational amplifiers $A_{1}$ and $A_{2}$ can be compensated by the current flowing into the operational amplifiers $A_{3}$ and $A_{4}$, i.e.,(27)$$(i_{A1}+i_{A2})+(i_{A3}+i_{A4})=0$$

Therefore, based on the discussions above, the Riordan floating simulated inductor can be regarded as an independent module like a practical inductor. That is convenient for practical applications. However, the electronic components in the Riordan floating simulated inductor are required to be strictly equal to their mirror components. The circuit structure of the Riordan floating simulated inductor is relatively complex and it includes four closed-loops, so, its realization needs higher quality components and higher circuit design skill. Meanwhile, the stability of the Riordan floating simulated inductor should be seriously considered, because it is a multi-closed-loop system.

Based on the above analysis and discussions, it can be found that the two simulated inductors both have their advantages and some limitations or preconditions for practical applications. Therefore, up to date, the simulated inductor technique is mainly studied and applied in the field of electronic communications, for either the Riordan grounded simulated inductor or the Riordan floating simulated inductor.

## 3. The Requirement of the Inductor Module in CET System

In the field of process tomography, different kinds of CET systems have been studied, including the capacitively coupled electrical resistance tomography (CCERT) system, the capacitively coupled electrical impedance tomography (CCEIT) system and the electrical capacitance tomography (ECT) system [18,19,20]. Figure 2a shows a sketch of a 12-electrode CET sensor. The electrodes of the CET sensor are mounted symmetrically around the outer surface of the insulating pipe. Figure 2b shows the equivalent circuit of an electrode pair in the CET sensor. It is simplified as two coupling capacitances $C_{1}$ and $C_{2}$ (formed by the two electrodes, the insulating pipe, and the measured fluid), and an impedance $Z_{x}$ of the measured fluid. For the CCERT system, the measured fluid is equivalent to a resistance $R_{x}$, i.e., $Z_{x}=R_{x}$. For the ECT system, the measured fluid is equivalent to a capacitance $C_{x}$, i.e., $Z_{x}=C_{x}$. For the CCEIT system, the measured fluid is equivalent to an impedance $Z_{x}$.

It is worth mentioning that the existence of the two coupling capacitances $C_{1}$ and $C_{2}$ makes the contactless measurement possible. However, from the viewpoint of electrical impedance measurement, only the $Z_{x}$ is the useful signal. The capacitive reactance of $C_{1}$ and $C_{2}$ is the background signal which will limit the signal-to-noise ratio (SNR) of impedance measurement and should be overcome.

Research works have verified that the impedance elimination principle can provide an effective approach to overcome the unfavorable influences of the coupling capacitances, i.e., by introducing an inductor module and using the inductive reactance of the inductor module to eliminate the capacitive reactance of the coupling capacitances. 

Figure 3 shows the flowchart of the impedance elimination principle for the measurement of an electrode pair in the CET sensor where f is the frequency of the excitation signal and j is the imaginary unit. The overall impedance Z of the measurement path is:(28)$$Z=\frac{1}{j2\pi fC_{1}}+Z_{x}+\frac{1}{j2\pi fC_{2}}+j2\pi fL$$

With the application of the impedance elimination principle, the imaginary part of the Z should be zero, i.e.:(29)$$\frac{1}{j2\pi fC_{1}}+\frac{1}{j2\pi fC_{2}}+j2\pi fL=0$$

Thus, from Equation (29), the excitation frequency f. is determined by:
(30)$$f=\frac{1}{2\pi }\sqrt{\frac{C_{1}+C_{2}}{LC_{1}C_{2}}}$$

Equations (28) to (30) show that if the excitation frequency is set by Equation (30), the impedance of the measurement path only consists of the impedance of measured fluid $Z_{x}$, i.e.:(31)$$Z=Z_{x}$$

Therefore, the capacitive reactance of the coupling capacitances can be successfully eliminated by the inductive reactance of the introduced inductor module. Thus, the negative influence of the background signal (coupling capacitances $C_{1}$ and $C_{2}$) can be effectively overcome. The SNR of the CET system can be improved.

According to Equation (30), it can be found that the system excitation frequency f is related to the inductance value L of the introduced inductor module and the coupling capacitances $C_{1}$ and $C_{2}$.

In practical application, the value of the excitation frequency f shouldn’t be set too large in order to avoid imposing extra high requirements on the circuit design and system complexity. Meanwhile, the values of the coupling capacitances are usually small (about 30~50 pF), so the introduced inductor module is expected to have a large inductance value.

Further, the materials, the wall thickness and the diameters of the insulating pipes as well as the electrode angle and the length of the electrode, etc., will also affect the values of the coupling capacitances, so the inductance value L of the inductor module should be adjustable and the adjustment range is desired to be as wide as possible. That means the inductor module should have good compatibility and adaptability.

Besides, the CET system is required to be compacted for miniaturization and integration. So the introduced inductor module should also have the capability to be minimized and can be easily integrated in the CET system.

Based on the above considerations, the problems have clearly indicated that the practical inductor still cannot satisfy the requirements of the practical application in the CET system, even if it is a simple way to use a practical inductor to form the inductor module.

Obviously, the simulated inductor technique provides an attractive approach to develop the inductor module and hence effectively overcome the unfavorable influences of the coupling capacitances. However, in the research field of CET, the applications and research of the simulated inductor are not sufficient. More research work should be undertaken.

## 4. New Simulated Inductor Module

### 4.1. The Design of the New Simulated Inductor Module

This work aims to develop a new simulated inductor which is suitable for the CET systems. Based on the discussions in Section 2 and Section 3, the new simulated inductor module should have the following features:(1)The new simulated inductor module should have a simple circuit construction.(2)The new simulated inductor module should be regarded as an independent module and can be connected into the circuit flexibly.(3)The input current of the new simulated inductor should be equal to the output current, because the CET system implements the impedance measurements by measuring the current flowing through the detection path, as shown in Figure 3.

Figure 4 illustrates the circuit of the new simulated inductor module.

In Figure 4, the new simulated inductor consists of a Riordan grounded simulated inductor with an independent power source and an I/V (current to voltage) convertor. Compared with the typical Riordan grounded simulated inductor in Figure 1a, the new simulated inductor has two key improvements (or differences):
(1)The output terminal (marked as $u_{out}$) is not connected to the ground directly. Instead, its output terminal is connected to the inverting input terminal of the operational amplifier of the I/V converter.(2)The operational amplifiers in the Riordan grounded simulated inductor are supplied by an independent power source, while other operational amplifiers are powered by the system power source. Meanwhile, the common node ($u_{com}$) of the independent power is connected to the output terminal.

From Section 2, the output terminal of the Riordan grounded simulated inductor needs to be grounded, to ensure that the simulated inductor module has a fixed equivalent inductance value $L_{eq}$ (as discussed in Section 2.1, the pre-condition to obtain a fixed value of $L_{eq}$ is that the load of the circuit is fixed). However, in practical applications, changes of the succeeding circuit should not affect the inductance value $L_{eq}$ of an independent inductor module. Therefore, as shown in Figure 4, an equivalent grounded method is introduced to meet the application pre-condition of the Riordan grounded simulated inductor, i.e., connecting the output terminal to the inverting input terminal of an operational amplifier, and the noninverting terminal of the operational amplifier is directly connected to the ground. Meanwhile, as shown in Figure 4, the detection current signal $i_{out}$ which flows through the new simulated inductor is transmitted to a output voltage signal $u_{f}$ by the I/V convertor.

Additionally, according to the discussion in Section 2.1, since the currents flow into the operational amplifiers ($A_{1}$ and $A_{2}$) in Figure 1a, the output current of the Riordan grounded simulated inductor is different from the input current, so the Riordan grounded simulated inductor can’t be used as an independent inductor. To solve this problem, we should seek an effective way to make the input current equal to the output current. Our method is to compensate the currents, which flow into the operational amplifiers, to main circuit.

Figure 5 shows the operation characteristics of a typical operational amplifier [1,33,34]. In Figure 5, $u_{com}$ is the common node of the independent power source. $i_{n}$ is the current into the inverting input terminal. $i_{p}$ is the current into the noninverting input terminal. $i_{o}$ is the current into the output terminal. $i_{c+}$ is the current flowing from the positive power supply terminals to common node. $i_{c-}$ is the current flowing from the negative power supply terminal to common node. $i_{c}$ is the current obtained from the common node.

The sum of the currents entering the operational amplifier should be zero:(32)$$i_{p}+i_{n}+i_{o}-\left(i_{c+}+i_{c-}\right)=0.$$

Meanwhile, there is no current flowing into either input terminal (i.e., $i_{p}=i_{n}=0$). The current $i_{o}$ flowing into the output terminal equals to the sum of the currents $i_{c+}$ and $i_{c-}$. The current obtained from the common node $u_{com}$ is also the sum of the currents $i_{c+}$ and $i_{c-}$. Thus, the following equation can be obtained:(33)$$i_{o}=i_{c+}+i_{c-}=i_{c}$$

Equation (33) indicates that the current which flows into the output terminal of operational amplifier equals to the current which is obtained from the common node of the independent power source. Obviously, it is independent on the power sources of other operational amplifiers.

In the case of this work, two operational amplifiers $A_{1}$ and $A_{2}$ are used in the grounded simulated inductor as shown in Figure 5. $i_{A1}$ is the current into the output terminal of $A_{1}$. $i_{A2}$ is the current into the output terminal of $A_{2}$. As mentioned in Section 2.1, the input current of a grounded simulated inductor is not equal to the output current. There exist some differences and the differences between the input current and the output current are $i_{A1}$ and $i_{A2}$ (as shown in Figure 1). In another respect, based on the above discussion in this section, the current which flows into the output terminal of an operational amplifier can be obtained from the common node of the independent power source. Therefore, making the input current equal to the output current can be realized by introducing the current from the common node of the independent power source into the main circuit.

The new simulated inductor module has two currents ($i_{A1}$ and $i_{A2}$) which should be compensated. Simply, we can use two independent power sources and introduce the currents from the two common nodes of the two independent power sources. It is a direct treatment. But, it needs two independent power sources. In fact, in this work, it is not necessary to use two independent power sources, only one independent power source can also meet the requirement, because the currents ($i_{A1}$ and $i_{A2}$) can only flow from the common node of the independent power source. There is no other way. So, in this work, the two operational amplifiers share one independent power source which is independent of the power sources of other operational amplifiers. Thus, our method to make the input current equal to the output current is implemented by introducing the current from the common node into the main circuit. As shown in Figure 4, $i_{A1}$ and $i_{A2}$ can be obtained from one common node $u_{com}$:(34)$$i_{com}=i_{A1}+i_{A2}$$

Further, the current $i_{out}$ is:(35)$$i_{out}=i_{R5}+i_{com}=i_{R5}+(i_{A1}+i_{A2})=i_{in}$$

From the above discussions, it can be found that:
(1)By connecting the output terminal to the inverting input terminal of the operational amplifier, equivalent grounded is realized, which meets the requirement of pre-condition of the grounded simulated inductor. The equivalent inductance *L_eq_* of the new simulated inductor module is guaranteed to be fixed. It is not affected by the load of the succeeding circuit.(2)By connecting the common node of the independent power source to the main circuit, the currents which flow into the output terminals of the operational amplifiers $A_{1}$ and $A_{2}$ are compensated. The input current of the new simulated inductor module equals to the output current ($i_{in}=i_{out}$). Further, the input impedance $Z_{in}$ of the new simulated inductor equals to its output impedance $Z_{out}$.

Therefore, the new simulated inductor not only has the simple construction and good stability similar to that of the grounded simulated inductor, but also can be used conveniently in practical applications like the floating simulated inductor. It reduces the requirements for components and can be used as an independent inductor module to effectively replace the practical inductor. Correspondingly, the equivalent inductance. *L_eq_* and the internal resistance $r_{eq}$ of the new simulated inductor module are:(36)$$L_{eq}=\frac{C_{m}R_{3}R_{5}}{R_{2}}R_{1}$$
(37)$$r_{eq}=\frac{R_{3}R_{5}}{R_{2}R_{4}}R_{1}$$

Comparing Equations (36) and (37) with Equations (11) and (12), it can be found that the $L_{eq}$ and the $r_{eq}$ of the new simulated inductor are the same as those of the grounded simulated inductor.

### 4.2. Impedance Measurement Principle by Using the New Simulated Inductor Module

Figure 6 shows the simplified measurement principle circuit of the CET system by using the new simulated inductor module, where the impedance of the measured fluid is $Z_{x}$.

The overall impedance Z of the measurement path is:(38)$$Z=\frac{1}{j2\pi fC_{1}}+Z_{x}+\frac{1}{j2\pi fC_{2}}+r_{eq}+j2\pi fL_{eq}$$

The excitation frequency *f* can be determined by the following equations:(39)$$\frac{1}{j2\pi fC_{1}}+\frac{1}{j2\pi fC_{2}}+j2\pi fL_{eq}=0$$
(40)$$f=\frac{1}{2\pi }\sqrt{\frac{C_{1}+C_{2}}{L_{eq}C_{1}C_{2}}}$$

At the excitation frequency, the capacitance reactance of $C_{1}$ and $C_{2}$ is eliminated by the inductive reactance of $L_{eq}$. The unfavorable influences of the coupling capacitances are overcome by the impedance elimination principle. As a result, the overall impedance of the measurement path Z consists of the impedance of measured fluid $Z_{x}$ and the internal resistance $r_{eq}$, i.e.:(41)$$Z=Z_{x}+r_{eq}$$

After the operation of the I/V convertor, the output voltage signal $u_{f}$ is:(42)$$u_{f}=-i_{out}R_{f}=-\frac{u}{Z_{x}+r_{eq}}R_{f}$$

From Equation (37), it can be seen that if $R_{4}$ is selected as a large-valued resistor, the value of $r_{eq}$ can be very small compared to the impedance of the measured fluid $Z_{x}$. Thus, Equation (40) can be further simplified as:(43)$$u_{f}\approx -\frac{u}{Z_{x}}R_{f}$$

Equation (43) indicates that the phase difference between u and $u_{f}$ becomes 180° when the impedance elimination principle is realized. In practical measurement process, it is not necessary to know the exact value of $C_{1}$ or $C_{2}$. The excitation frequency can be predetermined by an oscilloscope.

In the case when the independent power source is not available, the combination of the Riordan grounded simulated inductor and the I/V convertor can also construct a simulated inductor module as shown in Figure 7. The simulated inductor module can also be workable. It has the same equivalent inductance *L_eq_* and internal resistance $r_{eq}$ as those of the new simulated inductor module, but there are two problems:
(1)The input current is not equal to the output current ($i_{in}\neq i_{out}$).(2)When the impedance elimination principle is realized, the phase difference between the input current $i_{in}$ and the output current $i_{out}$ is approximately 90° rather than 180°.

In this case, as discussion in Section 2.1, the input current $i_{in}$ is not equal to the output current $i_{out}$. The output voltage signal $u_{f}$ is:(44)$$u_{f}=-i_{out}R_{f}$$

Meanwhile, according to Figure 7 and the discussions in Section 2.1, the following equations can be obtained:(45)$$i_{in}=\frac{u-u_{in}}{\frac{1}{j2\pi fC_{1}}+Z_{x}+\frac{1}{j2\pi fC_{2}}}$$
(46)$$i_{out}=\frac{\left(1/R_{4}+j2\pi fC_{m}\right)R_{1}R_{3}}{R_{2}}i_{in}=\frac{u_{in}}{R_{5}}$$
(47)$$i_{out}=\frac{\left(1/R_{4}+j2\pi fC_{m}\right)R_{1}R_{3}}{R_{2}}\cdot \frac{u}{\frac{1}{j2\pi fC_{1}}+Z_{x}+\frac{1}{j2\pi fC_{2}}+\frac{\left(1/R_{4}+j2\pi fC_{m}\right)R_{1}R_{3}}{R_{2}}\cdot R_{5}}$$
(48)$$i_{out}=\frac{r_{eq}+j2\pi fL_{eq}}{R_{5}}\cdot \frac{u}{\frac{1}{j2\pi fC_{1}}+Z_{x}+\frac{1}{j2\pi fC_{2}}+r_{eq}+j2\pi fL_{eq}}$$

When the impedance elimination principle is realized, Equation (48) is simplified as:(49)$$i_{out}=\frac{r_{eq}+j2\pi fL_{eq}}{R_{5}}\cdot \frac{u}{Z_{x}+r_{eq}}$$

Thus:(50)$$u_{f}=-\frac{r_{eq}+j2\pi fL_{eq}}{R_{5}}\cdot \frac{u}{Z_{x}+r_{eq}}R_{f}$$

Further, when the value of $r_{eq}$ is relatively small to the impedance of measured fluid $Z_{x}$, Equation (50) can be further simplified as:(51)$$u_{f}\approx -j\frac{2\pi fL_{eq}}{R_{5}}\cdot \frac{u}{Z_{x}}R_{f}$$

Therefore in this case, from Equation (51), the phase difference between the excitation signal u and the output voltage signal $u_{f}$ is approximately 90° when the impedance elimination principle is realized.

Based on the above discussion, the new simulated inductor module is also workable in the case when an independent power source is not available. It is also an effective alternative method. However, it is necessary to indicate that, in this case, the input current of the simulated inductor module is not equal to the output current ($i_{in}\neq i_{out}$) and the phase difference between input signal and output signal is approximately 90°. That will more or less have influence on the measurement results.

## 5. Experimental Results and Discussion

To test the effectiveness and evaluate the performance of the new simulated inductor with independent power source in the CET system, two sets of experiments were carried out: impedance measurement experiments simulating the characteristics of the measured fluid (hereinafter referred to as simulation experiments), and image reconstruction experiments based on a CET system using the classic linear back projection (LBP) algorithm. More detailed information on the construction of the CET system can be found in [32] ([32] focuses on the hardware improvement of the CCERT system which used the simulated inductor technique. But the detailed information of the simulated inductor module are not discussed and provided in [32]).

The components information of the new simulated inductors (Figure 4) was: the value of the adjustable resistor $R_{1}$ ranged from 0 to 10.0 kΩ, $R_{2}$ = $R_{3}$ = 3.30 kΩ, $R_{4}$ = 1.00 MΩ, $R_{5}$ = 5.10 kΩ, and $C_{m}$ = 2.20 nF. The operational amplifiers ($A_{1}$ and $A_{2}$) were AD817 (Analog Devices, Inc., Norwood, MA, USA). The operational amplifier ($A_{3}$) of the I/V converter was LM6172 (Texas Instruments, Inc., Dallas, TX, USA). The feedback resistor *R_f_* was 200 Ω.

### 5.1. The Simulation Experiments

In the simulation experiments, the equivalent coupling capacitances of the CET system were represented by two capacitors $C_{1s}$ and $C_{2s}$ with suitable fixed values. The practical measured fluid was represented by RC series combinations $Z_{x}$ with different impedance values.

The simulation experimental setup is shown in Figure 8.

In the simulated measurement path, the true values of the capacitor $C_{1s}$ and $C_{2s}$ were 9.97 pF and 9.99 pF, respectively. Correspondingly, the equivalent inductance value of new simulated inductor module was adjusted to 81.23 mH. Three groups of resistors, capacitors, and their combinations were measured:(1)Group 1 used some different resistors to simulate the ERT system, the true values of the resistors were 10.06 kΩ, 20.16 kΩ, 30.42 kΩ, 39.64 kΩ, 47.58 kΩ, 57.73 kΩ, 76.65 kΩ, 83.30 kΩ and 100.36 kΩ.(2)Group 2 used some different capacitors to simulate ECT system, the true values of the capacitors were 2.12 pF, 5.52 pF, 9.97 pF, 15.32 pF and 33.87 pF.(3)Group 3 used some RC series combinations to simulate the EIT system, six kinds of RC series combinations were formed by resistors of 20.16 kΩ, 47.58 kΩ, 76.65 kΩ and capacitors of 5.52 pF, 15.32 pF.

The true values of the resistors and the capacitors were calibrated by a commercial impedance analyzer (Keysight 4294A, Santa Rosa, CA, USA, 40 Hz to 110 MHz) at 250 kHz. The AC excitation signal was generated by a digital signal generator. The AC excitation signal had the amplitude value of 1.52 V and its frequency is 250 kHz. The total impedance information was measured by the signal processing module.

To test the performance of the measurement, four performance indexes are selected, including $e_{R}$, $e_{C}$, $\sigma_{R}$ and $\sigma_{C}$. The $e_{R}$ and $e_{C}$ are the relative errors (%) of the resistance measurement and the capacitance measurement. $\sigma_{R}$ and $\sigma_{C}$ are the standard deviations of the resistance measurement and the capacitance measurement, respectively. The four performance indexes are defined as follows:(52)$$e_{R}=\frac{R}{R_{r}}-1$$
(53)$$e_{C}=\frac{C}{C_{r}}-1$$
(54)$$\sigma_{R}=\sqrt{\frac{1}{K-1}\overset{K}{\underset{k=1}{\sum }}{\left(R_{k}-\frac{1}{K}\overset{K}{\underset{k=1}{\sum }}R_{k}\right)}^{2}}$$
(55)$$\sigma_{C}=\sqrt{\frac{1}{K-1}\overset{K}{\underset{k=1}{\sum }}{\left(C_{k}-\frac{1}{K}\overset{K}{\underset{k=1}{\sum }}C_{k}\right)}^{2}}$$
where $R_{m}$ and $C_{m}$ are the mean values of the measurement results of the resistance and the capacitance of the impedance, $R_{m}=\frac{1}{K}\sum_{k=1}^{K}R_{k}$ and $C_{m}=\frac{1}{K}\sum_{k=1}^{K}C_{k}$. $R_{r}$ and $C_{r}$ are the true values of the resistance and the capacitance. $R_{k}$ and $C_{k}$ are the *k*th measurement results of the resistance and the capacitance. *K* is the total number of repeated measurements (in this work, *K* = 100).

Figure 9a,b shows the measurement results of Group 1 and Group 2, respectively. The maximum relative error of the resistance measurement $e_{R}$ was 2.15%. The maximum relative error of the capacitance measurement $e_{C}$ was 2.99%. The maximum standard deviation of the resistance measurement was 0.50 kΩ. The standard deviation of the capacitance measurement was 0.05 pF.

Table 1 shows the experimental results of the impedance measurement experiments (Group 3). From Table 1, it can be seen that the CET system based on the new simulated inductor module can effectively obtain the total impedance information. The maximum relative error of the impedance measurement of RC series combination was 4.77%. The maximum standard deviation of the resistance measurement was 0.27 kΩ. The maximum standard deviation of the capacitance measurement was 0.07 pF.

Correspondingly, in the case when the independent power source of the new simulated inductor module is not available, the impedance measurement experiments were also carried out. The measurement results of Group 1, Group 2 and Group 3 are shown in Figure 10a, Figure 10b and Table 2, respectively. In this case, the maximum relative error of the resistance measurement $e_{R}$ was 2.66%. The maximum relative error of the capacitance measurement $e_{C}$ was 3.13%. The maximum relative error of the impedance measurement of RC series combination was 4.84%. The maximum standard deviation of the resistance measurement was 0.43 kΩ. The maximum standard deviation of the capacitance measurement was 0.08 pF.

The simulated experimental results have demonstrated that the design of the new simulated inductor module is effective, and it can replace the practical inductor to achieve impedance elimination in the measurement path of the CET system, and its performance was satisfactory. The unfavorable influences of coupling capacitances can be successfully overcome by using the new simulated inductor modules, and the SNR of the CET system is improved.

Meanwhile, it is also found that the performance of the new simulated inductor without the independent power source is comparable to that of the new simulated inductor with the independent power source. There only exists slight difference between the experimental results of these two cases. So in the case when the independent power source is not available, the new simulated inductor is also an effective alternative method, but the phase difference between input signal and output signal is approximately 90° when the elimination principle is realized.

### 5.2. The Practical Image Reconstruction Experiments

To test the image reconstruction quality of the CET system using new simulated inductors modules, the practical image reconstruction experiments were carried out. The material of insulating pipe of the CET sensor was polyvinyl chloride (PVC). The outer diameter and the thickness of the PVC pipe were 110 mm and 2 mm, respectively. The length of electrode and the electrode angel were 150 mm and 26°, respectively. Figure 11 is a photo of the CET system prototype.

Tap water with the conductivity of 160 µS/cm was used as the continuous phase. Non-conductive plastic (polyethylene) rods with different diameters (20 mm and 35 mm) were used as the discrete phase. During the experiments, the plastic rods were put into different positions in the pipe which was full of tap water. Then, the conductivity changes in the pipe can be detected. The classic LBP algorithm was used to implement the image reconstruction.

In order to compare the image reproduction effect, three image quality indexes were used [21]: mean squared error (MSE), relative image error (RIE) and image correlation coefficient (ICC). MSE is evaluated according to Equation (56). RIE is evaluated according to Equation (57). ICC is evaluated according to Equation (58):(56)$$MSE=\frac{1}{N}\sum_{n=1}^{N}{\left(\hat{g}-g\right)}^{2}$$
(57)$$RIE=\frac{\left\|\hat{g}-g\right\|}{\left\|g\right\|}$$
(58)$$ICC=\frac{\sum_{n=1}^{N}\left({\hat{g}}_{n}-\bar{\hat{g}}\right)\left(g_{n}-\bar{g}\right)}{\sqrt{\sum_{n=1}^{N}{\left({\hat{g}}_{n}-\bar{\hat{g}}\right)}^{2}\sum_{n=1}^{N}{\left(g_{n}-\bar{g}\right)}^{2}}}$$
where, ${\hat{g}}_{n}$ is the gray level of the *n*th pixel of the reconstructed images and $\hat{g}={\left[{\hat{g}}_{1},{\hat{g}}_{2}\ldots {\hat{g}}_{n}\ldots {\hat{g}}_{N}\right]}^{T}$ is the vector of the gray levels of the reconstructed images. $g_{n}$ is the practical gray level of the *n*th pixel of the conductivity distributions and $g={\left[g_{1},g_{2}\ldots g_{n}\ldots g_{N}\right]}^{T}$ is the vector of the gray levels of the practical conductivity distributions. $\bar{\hat{g}}$ and $\bar{g}$ are the mean value of $\hat{g}$ and g respectively. *N* is the number of the pixel of the reconstructed images, in this work, *N* = 856.

Figure 12 is the experimental results of image reconstruction. Table 3 lists the information of the three image quality indexes. 

The practical experimental image reconstruction results indicate that the quality of the image reconstruction of the CET system prototype using new simulated inductors modules is satisfactory. Compared with the CET system without inductor module, the MSE and RIE of the CET system using new simulated inductor module are smaller. The quality of the image reconstruction of CET system is improved by introducing the new simulated inductor module. The application of simulated inductor technique to the research field of the process tomography is effective.

## 6. Conclusions

This work analyzes the characteristics of the simulated inductor based on the Riordan circuit in detail. Through comparison and analysis, it is found that the grounded simulated inductor has a simple circuit construction but one terminal should be grounded. The floating simulated inductor can be used as an independent inductor module while its circuit structure is relatively complex. Meanwhile, it is also pointed out that, the reason why the grounded simulated inductor can’t be used as an independent module is that the output current of the grounded simulated inductor is different from the input current ($i_{in}\neq i_{out}$).

A new simulated inductor module which is suitable for CET system is reported in this work. By compensating the currents from the common node of the introduced independent power source to the main circuit, a new simulated inductor is designed. The new simulated inductor not only has a simple construction similar to the grounded simulated inductor, but also can be used as an independent module like the floating simulated inductor. It reduces the requirements for components and can be used conveniently in practical applications like a practical inductor. The new simulated inductor module guarantees the equivalence between the input current and the output current. These two advantages make it an attractive approach to apply the simulated inductor technique to the research fields of process tomography.

To test the effectiveness and evaluate the performance of the new designed simulated inductor, two sets of experiments were carried out. The experimental results show that the design of the new simulated inductor is successful and the performance of the new simulated inductor is satisfactory. The maximum relative error of the resistances measurement is 2.15%, the maximum relative error of the capacitances measurement is 2.99%, and the maximum relative error of the impedance measurement is 4.77%. The practical image reconstruction of the CET system results indicate that the quality of the image reconstruction is satisfactory and the application of simulated inductor technique to the research field of the process tomography is effective. The SNR of the CET system can be improved.

Besides, the experimental results also indicate that, in the case when the independent power source is not available, the performance of the new simulated inductor is comparable to that of the new simulated inductor with the independent power source. So in this case, the new simulated inductor is also an effective alternative method, but the phase difference between input signal and output signal is approximately 90° when the elimination principle is realized (the input current is not equal to the output current). That may more or less have influence on measurement results and practical application.

This work provides the detailed information on simulated inductors based on Riordan circuit, including the detailed measurement circuits and related analysis, the technical parameters of the components of the simulated inductor, the obtained experience/knowledge and the latest progresses, as well as the design of a new simulated inductor. According to the technical content in the paper, readers can duplicate the measurement circuit of the new simulated inductor. We hope that this paper can provide useful reference/experience for other researchers’ work and extend the application fields of the simulated inductor technique.

## Figures and Tables

**Figure 1 sensors-19-02463-f001:**
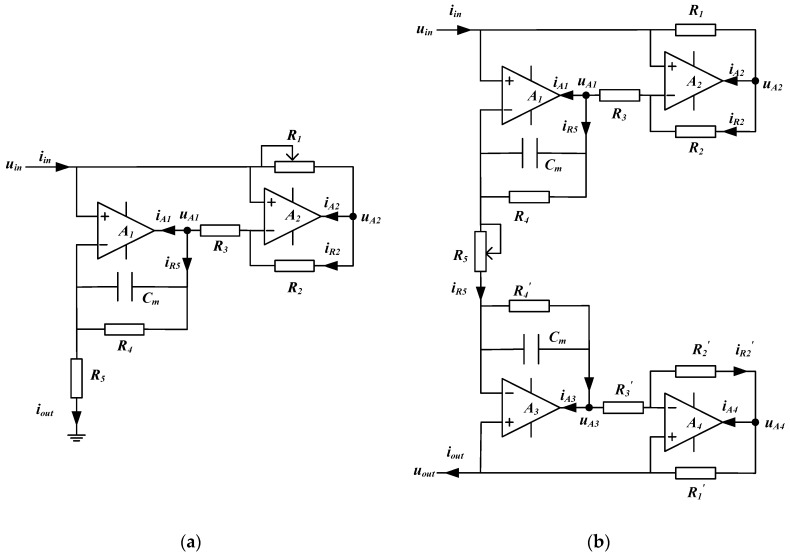
The application circuits of two types Riordan simulated inductors: (**a**) the Riordan grounded simulated inductor; (**b**) the Riordan floating simulated inductor.

**Figure 2 sensors-19-02463-f002:**
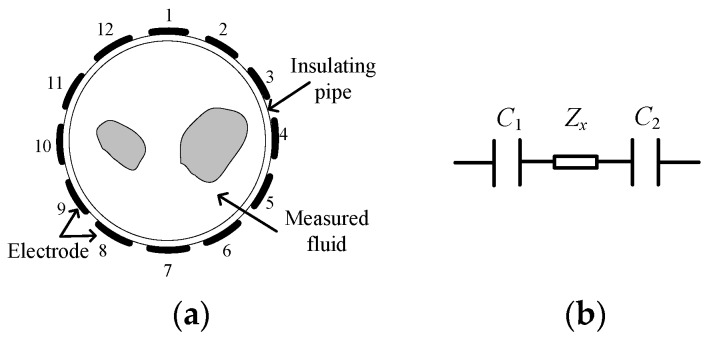
Measurement principle of the CET sensor. (**a**) The sketch of a 12-electrode CET sensor; (**b**) The equivalent circuit of a measurement electrode pair in the CET sensor.

**Figure 3 sensors-19-02463-f003:**
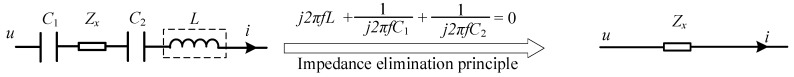
The flowchart of the impedance elimination principle in the CET systems.

**Figure 4 sensors-19-02463-f004:**
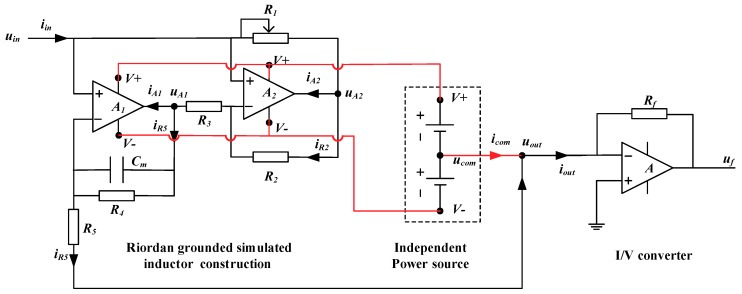
The circuit of the new simulated inductor module.

**Figure 5 sensors-19-02463-f005:**
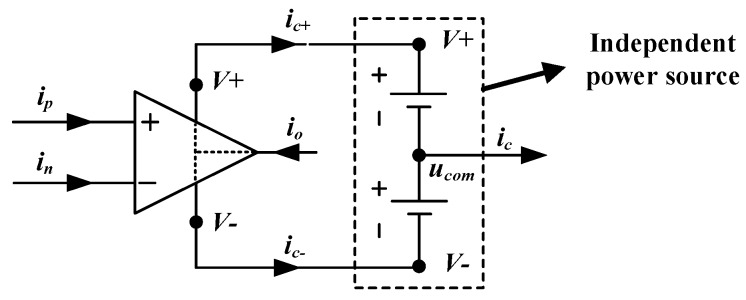
The operation characteristics of a typical operational amplifier.

**Figure 6 sensors-19-02463-f006:**
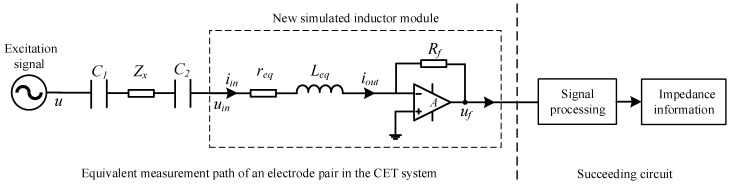
The simplified measurement principle circuit of the CET system.

**Figure 7 sensors-19-02463-f007:**
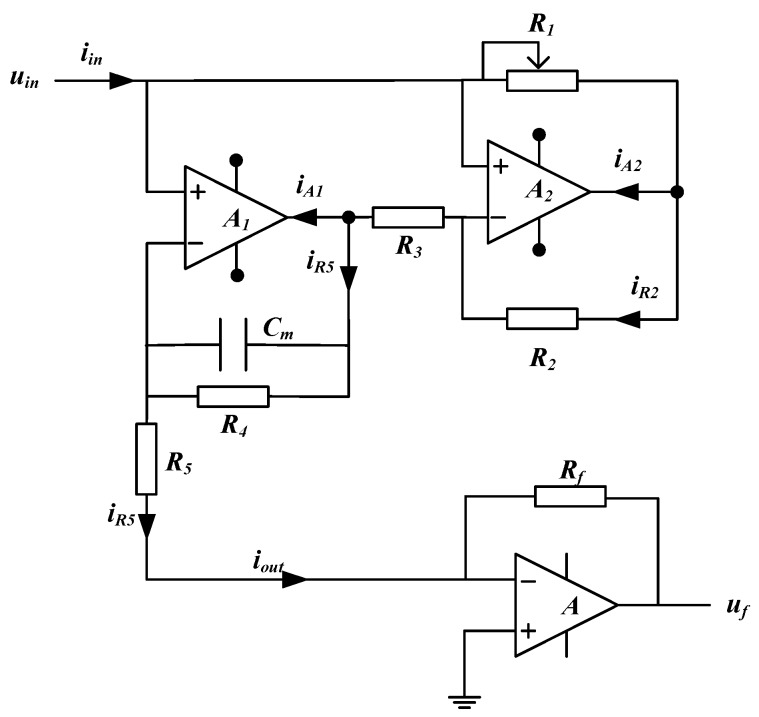
The simulated inductor module based on a Riordan grounded simulated inductor and a I/V convertor.

**Figure 8 sensors-19-02463-f008:**
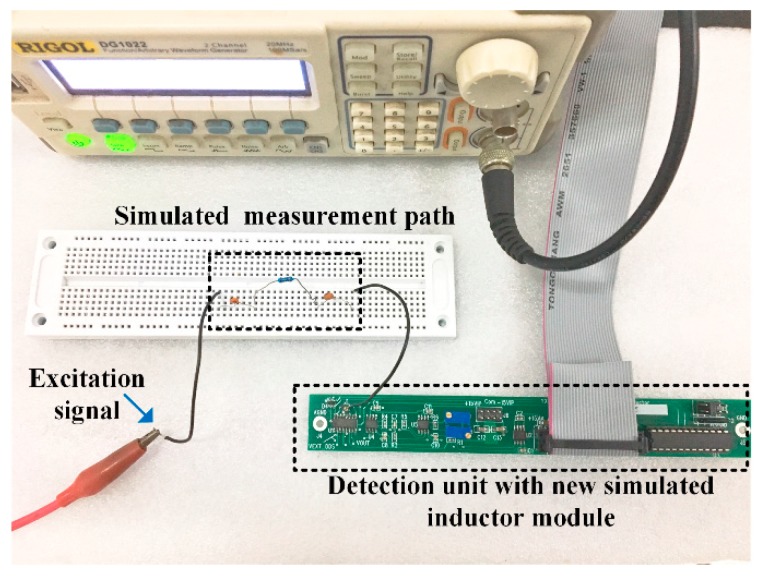
The simulation experimental setup.

**Figure 9 sensors-19-02463-f009:**
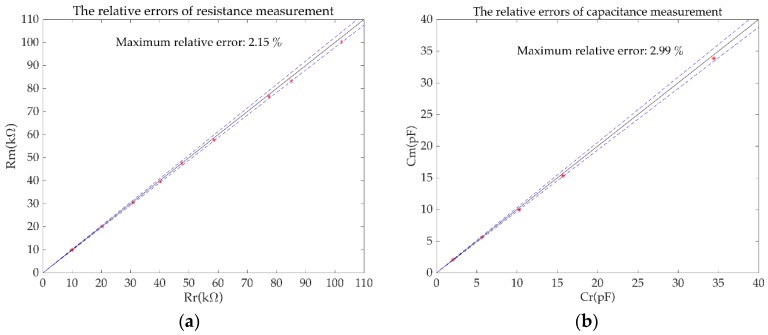
The measurement results of the simulation experiments. (**a**) the relative errors of resistance measurement; (**b**) the relative errors of capacitance measurement.

**Figure 10 sensors-19-02463-f010:**
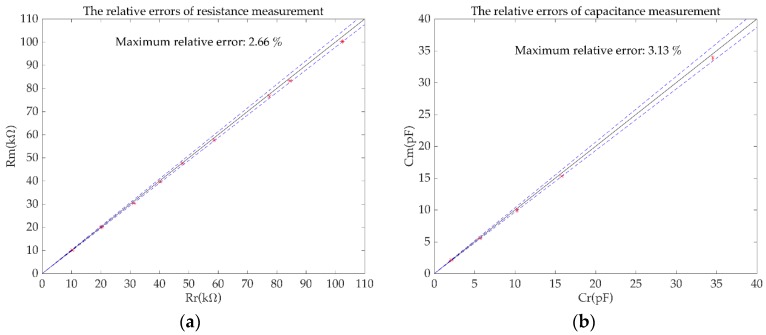
The measurement results of the simulation experiments without independent power source. (**a**) the relative errors of resistance measurement; (**b**) the relative errors of capacitance measurement.

**Figure 11 sensors-19-02463-f011:**
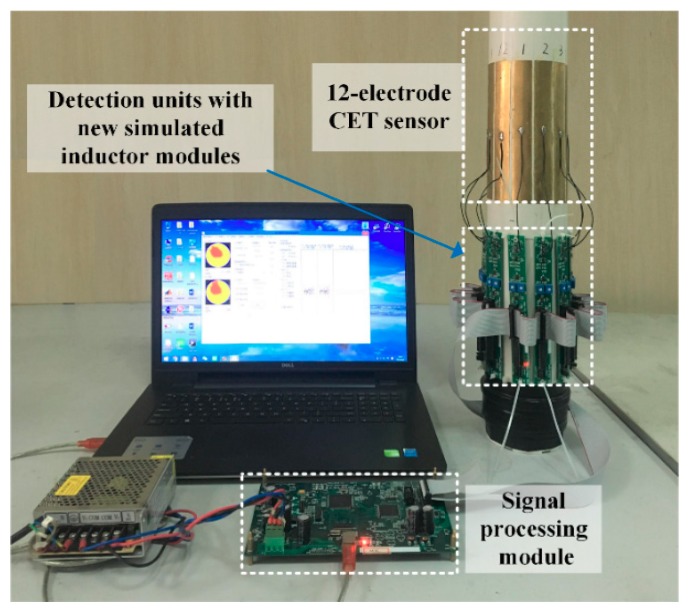
The photo of the CET system prototype.

**Figure 12 sensors-19-02463-f012:**
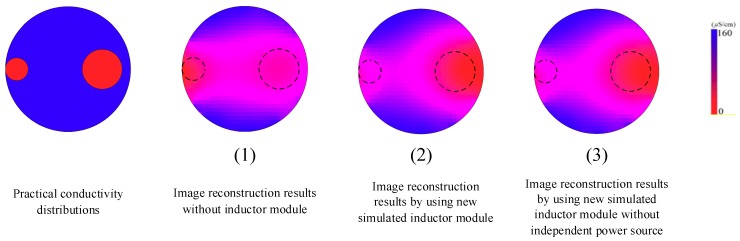
Practical conductivity distributions and image reconstruction results of CET systems.

**Table 1 sensors-19-02463-t001:** Impedance measurement results of the CET system with independent power source (250 kHz).

*R_r_* (kΩ)	*C_r_* (pF)	*R* (kΩ)	*C* (pF)	*e_R_* (%)	*e_C_* (%)	*σ_R_* (kΩ)	*σ_C_* (pF)
20.16	5.52	21.11	5.63	4.74	2.01	0.44	0.01
20.16	15.32	20.33	15.70	0.89	2.51	0.03	0.03
47.58	5.52	47.55	5.64	−0.04	2.20	0.20	0.02
47.58	15.32	46.98	15.74	−1.25	2.78	0.09	0.06
76.65	5.52	74.35	5.39	−2.99	−2.21	0.27	0.02
76.65	15.32	75.81	14.59	−1.09	−4.77	0.14	0.07

**Table 2 sensors-19-02463-t002:** Impedance measurement results of the CET system without independent power source (250 kHz).

*R_r_* (kΩ)	*C_r_* (pF)	*R* (kΩ)	*C* (pF)	*e_R_* (%)	*e_C_* (%)	*σ_R_* (kΩ)	*σ_C_* (pF)
20.16	5.52	20.81	5.60	3.26	1.47	0.43	0.01
20.16	15.32	20.83	15.84	3.37	3.45	0.03	0.03
47.58	5.52	47.43	5.61	−0.31	1.66	0.21	0.02
47.58	15.32	47.18	15.78	−0.83	3.04	0.08	0.07
76.65	5.52	74.47	5.45	−2.84	−1.12	0.30	0.02
76.65	15.32	75.52	14.57	−1.47	−4.84	0.14	0.08

**Table 3 sensors-19-02463-t003:** Three image quality indexes of image reconstruction results.

Image Reconstruction Results	(1)	(2)	(3)
MSE	0.0142	0.0138	0.0138
RIE	0.4805	0.4376	0.4408
ICC	0.5389	0.5514	0.5484

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
