# Peer review of "On the Design of a New Simulated Inductor Using a Contactless Electrical Tomography System as an Example"

_sensors, 2019, doi:10.3390/s19112463_

Round 1

Reviewer 1 Report

All comments are included in attached file.

Author Response

Reviewer 1

COMMENTS TO THE AUTHOR(S)

In manuscript authors present the detailed information and characteristics of inductors based on Riordan circuit. To practical application in CET systems authors propose usage the extension of grounded simulated inductor, which relies on introducing the independent power source into main circuit.

The article is prepared using commonly used standards (introduction with the state of art, Riordan inductor with some modifications, results and discussion). Nevertheless, to increase the value of manuscript, it is necessary to make a few corrections and explanations.

1. Punctuation marks should be corrected throughout the text;

R: As suggested, in the revised manuscript, punctuation marks have been checked and corrected.

2. In formula (2) the indexes should be improved as iin = iA2 + iR2;

R: Thanks for reviewer’s reminder. In the revised manuscript, the formula (2) has been corrected, which can be found in page 3.

3. Line 113, in sentence “The equivalent internal resistance Leq...” the notice Leq should be replaced by req.

R: Thank you for reviewer’s reminder. It is our carelessness. In the revised manuscript, it has been corrected, which can be found in line 116, page 4.

4. Line 119, “the value of req is mainly adjusted by changing the value of R4”. From figure 1a we see, that the resistance R4 is constant (not adjustable). Please explain it;

R: The reviewer is right, the resistance R4 is constant (not adjustable) in this work. It is our carelessness to make the manuscript confused. To obtain a small-valued internal resistance req, the resistance value of R4 is much greater than that of other resistances (R1 R2, R3and R5). In the revised manuscript, to avoid this confusion, the sentence has been changed to ‘the value of req is mainly determined by the value of R4’, which can be found in line 122, page 4.

5. Lines 152 -156, this same remark as presented above. On figure 1b for Riordan floating simulated inductor the resistant R5 is constant. Please explain how to adjust equivalent internal resistance req by changing resistance R5;

R: It is our carelessness to make the manuscript confused. To obtain a small-valued internal resistance req, the resistance value of R4 is much greater than that of other resistances (R1 R2, R3and R5). In the revised manuscript, corresponding content has been changed to ‘The value of req of the Riordan floating simulated inductor is mainly determined by the value of R4’, which can be found in line 155-156, page 5.

6. Lines 145 - 149, at the end of formula (21) please insert a full stop and write exactly, that by comparison formulas (11) - (12) and (20) - (21) respectively we can see …

R: As suggested, in the revised manuscript, the sentence ‘Compared with Equation (11) and (12), it can be found that… ’ have been changed to ‘By comparison Equations (11)-(12) and (20)-(21) respectively we can see that …’, which can be found in line 150, page 5.

7. The notations (about resistance, inductance, currents etc.) should be presented before calculations of properties for Riordan grounded and floating inductors. For example, in formula (1) we see the variable ω, which is not described in text at all. In formula (7) we have variable f, which denotes as frequency of excitation signal only just in line 214;

R: Thank you for reviewer’s reminder. As suggested, in the revised manuscript, formula (1) has been changed as Zin = uin/iin = j2πfLeq, where f  is the frequency of excitation signal, which can be found in line 94, page 3.

8. The sentences included in lines 327-328 and 331-332 are similar, it is an artificial extension of the text;

R: Thank you for reviewer’s reminder. As suggested, in the revised manuscript, the repetitive sentences (lines 331-332) have been deleted.

9. The formulas (11)-(12) and (36)-(37), which present inductance and resistance for Riordan grounded inductor and inductor with independent power source respectively, are identical. Additionally please compare the formulas (11), (20) and (36). In this case it’s enough only short description without duplicate the formulas.

R: Thank you for reviewer’s reminder. The formulas of inductance and resistance for Riordan grounded inductor ((11) and (12)) and inductor with independent power source ((36) and (37)) are identical indeed. However, to keep the systematization and readability of the manuscript, formulas (36)-(37) need to be retained. In the revised manuscript, we have clearly indicated that the Leq and the req of the new simulated inductor are same as those of the grounded simulated inductor, which can be found in lines 347-348, page 10-11. Thanks for your understanding!

10. Lines 402-405 “. . . please refer to the previous reports . . .”. If possible please to text a few sentences about main results obtained in [26];

R: Reference [26] (The new number is [32]) focuses on the hardware improvement of the CCERT system, which used the simulated inductor technique. But the detailed information of the simulated inductor module are not discussed and provided in reference [26]. As suggested, more descriptions about reference [26] have been added, which can be found in line 403-406, page 12.

11. In section 5 the authors present the comparison of resistance and capacitance errors for Riordan simulated inductor with independent power source and without it. If possible please attach some values of metrics e.g. Mean Squared Error (MSE), Relative Image Error (RIE), Image Correlation Coefficient (ICC) of comparison of image reconstruction presented in figure 11b.

R: Thank you for reviewer’s reminder. As suggested, in the revised manuscript, three quantitative indexes, including MSE, RIE and ICC, have been added, which can be found in line 504-512, page 16-17. Meanwhile, a new table (Table 3) and corresponding technical content has also been added, which can be found in line 517-524, page 17. 

Reviewer 2 Report

The simulated inductor or inductance is a well known problem, however it application to the contactless tomography seems to be interesting. Unfortunately there is not in the literatue some important papers . For example written by Frank Stefani, Thomas Gundrum, and Gunter Gerbe or the Polish authors dr. Tomasz Rymarczyk (for example Tomographic Imaging in Environmental, Industrial and Medical Applications published by Innovatio Publishing House, University of Economics and Innovation in Lublin, 2019) or works of Prof. S. Gratkowski team from West Pomeranian University of Technology.

I have some questions:

What does it means: "elimination principle" 

"As basic electronic elements, inductors are widely…" The inductors are rather electromagnetic devices than electronic one. So some comments here should be added.

In spite of rather poor results of imaging (with comparison of the results achieved by T. Rymarczyk) the paper is interesting but some conclusion with respect of much better results achieved so far should be added.

The paper in my opinion is worth of publication under one conditions. The literaturę should be supplemented.

Author Response

Reviewer 2

COMMENTS TO THE AUTHOR(S)

The simulated inductor or inductance is a well known problem, however it application to the contactless tomography seems to be interesting. Unfortunately there is not in the literatue some important papers. For example written by Frank Stefani, Thomas Gundrum, and Gunter Gerbe or the Polish authors dr. Tomasz Rymarczyk (for example Tomographic Imaging in Environmental, Industrial and Medical Applications published by Innovatio Publishing House, University of Economics and Innovation in Lublin, 2019) or works of Prof. S. Gratkowski team from West Pomeranian University of Technology.

R: Thank you for reviewer’s reminder and suggestions. As suggested, in the revised manuscript, six new related references [21-26] have been added, which can be found in line 601-613, page 19.

1. What does it means: "elimination principle" 

R: Impedance elimination principle means that the capacitive reactance of the capacitance can be eliminated by the inductive reactance of the introduced inductor module. In the revised manuscript, more detailed information concerning about ‘impedance elimination principle’ can be founded in line 206-224, page 7.

2. “As basic electronic elements, inductors are widely…” The inductors are rather electromagnetic devices than electronic one. So some comments here should be added.

R: Thanks for reviewer’s reminder. As suggested, in the revised manuscript, some statements have been corrected and some comments have been added, which can be founded in line 33-35, page 1.

3. In spite of rather poor results of imaging (with comparison of the results achieved by T. Rymarczyk) the paper is interesting but some conclusion with respect of much better results achieved so far should be added.

R: Thanks for reviewer’s reminder. The main purpose of this work is to verify the feasibility of the new simulated inductor module in Contactless Electrical Tomography (CET) system. As mentioned in the manuscript, under the same experimental condition (same image reconstruction algorithm and same distribution), the impedance measurement performance and the image reconstruction performance of the CET system with new simulated inductor module is better, which can be found in line 517-524, page 17. To seek more effective image reconstruction algorithm and hence to obtain higher quality reconstructed images of CET system is our future work.

Thanks for your understanding!

Finally, thanks for the editor’s and the reviewers’ comments and suggestions, which help us fill many holes and improve the technical quality of our manuscript.

Thanks for your help and time!

Best Regards.

Sincerely,

Prof. Zhiyao Huang

College of Control Science and Engineering

Zhejiang University

38, ZheDa Road,

Hangzhou, 310027

China

May 23, 2019
